# Co-expression of *Foxa.a*, *Foxd* and *Fgf9/16/20* defines a transient mesendoderm regulatory state in ascidian embryos

**Clare Hudson\*[†], Cathy Sirour, Hitoyoshi Yasuo\*[†]**

Laboratoire de Biologie du Développement de Villefranche-sur-mer, Observatoire Océanologique, Sorbonne Universités, UPMC Univ Paris 06, CNRS, Villefranche-sur-Mer, France

**Abstract** In many bilaterian embryos, nuclear β-catenin (nβ-catenin) promotes mesendoderm over ectoderm lineages. Although this is likely to represent an evolutionary ancient developmental process, the regulatory architecture of nβ-catenin-induced mesendoderm remains elusive in the majority of animals. Here, we show that, in ascidian embryos, three nβ-catenin transcriptional targets, *Foxa.a*, *Foxd* and *Fgf9/16/*20, are each required for the correct initiation of both the mesoderm and endoderm gene regulatory networks. Conversely, these three factors are sufficient, in combination, to produce a mesendoderm ground state that can be further programmed into mesoderm or endoderm lineages. Importantly, we show that the combinatorial activity of these three factors is sufficient to reprogramme developing ectoderm cells to mesendoderm. We conclude that in ascidian embryos, the transient mesendoderm regulatory state is defined by co-expression of *Foxa.a*, *Foxd* and *Fgf9/16/*20.

**\*For correspondence:** clare.
hudson@obs-vlfr.fr (CH); yasuo@
obs-vlfr.fr (HY)

[†]These authors contributed
equally to this work

**Competing interests:** The
authors declare that no
competing interests exist.

**Reviewing editor:** Alejandro
Sánchez Alvarado, Stowers
Institute for Medical Research,
United States

## Introduction

The mesoderm, endoderm and ectoderm arise during embryonic development by a process termed germ layer segregation. In many species, at least part of the endoderm and mesoderm derive from transient 'mesendoderm' precursors, as is the case in ascidian embryos (*Kimelman and Griffin, 2000*; *Rodaway and Patient, 2001*). However, the precise nature of this induced regulatory state is not well understood. In ascidians, the first animal-vegetal (A-V) oriented cell division generates the eight-cell stage embryo and segregates the mesendoderm and some neural lineages into two pairs of vegetal founder lineages (the A- and B-line) and the ectoderm (epidermis and neural) into two pairs of animal lineages (a- and b-line) (*Conklin, 1905*; *Nishida, 1987*). This study focuses on the A-line mesendoderm lineages. From the 8- to 16-cell stage, the two A4.1 blastomeres divide medio-laterally to generate the two pairs of neuro-mesendodermal NNE cells, for notochord/neural/endoderm (*Figure 1a*). NNE cells then divide along the A-V axis to generate NN cells (notochord/neural) and E cells (mostly endoderm) at the 32-cell stage (*Figure 1a*). Subsequently, NN cells segregate into notochord and neural lineages at the 64-cell stage. At this stage, while the medial E cell generates two endoderm precursors, the lateral-most E cell is subject to an inductive interaction resulting in the generation of one endoderm and one mesoderm (the trunk lateral cell lineage) precursor (*Shi and Levine, 2008*). Later, during neural plate patterning, a muscle precursor is also generated from the lateral borders of the NN-lineage-derived neural plate (*Nicol and Meinertzhagen, 1988*; *Nishida, 1987*). Thus, as in other species, ascidian germ layer segregation can be viewed as a progressive process with part of the neural tissue arising from bipotential neuro-mesodermal progenitors (*Henrique et al., 2015*; *Tzouanacou et al., 2009*). The earliest cell divisions of the ascidian

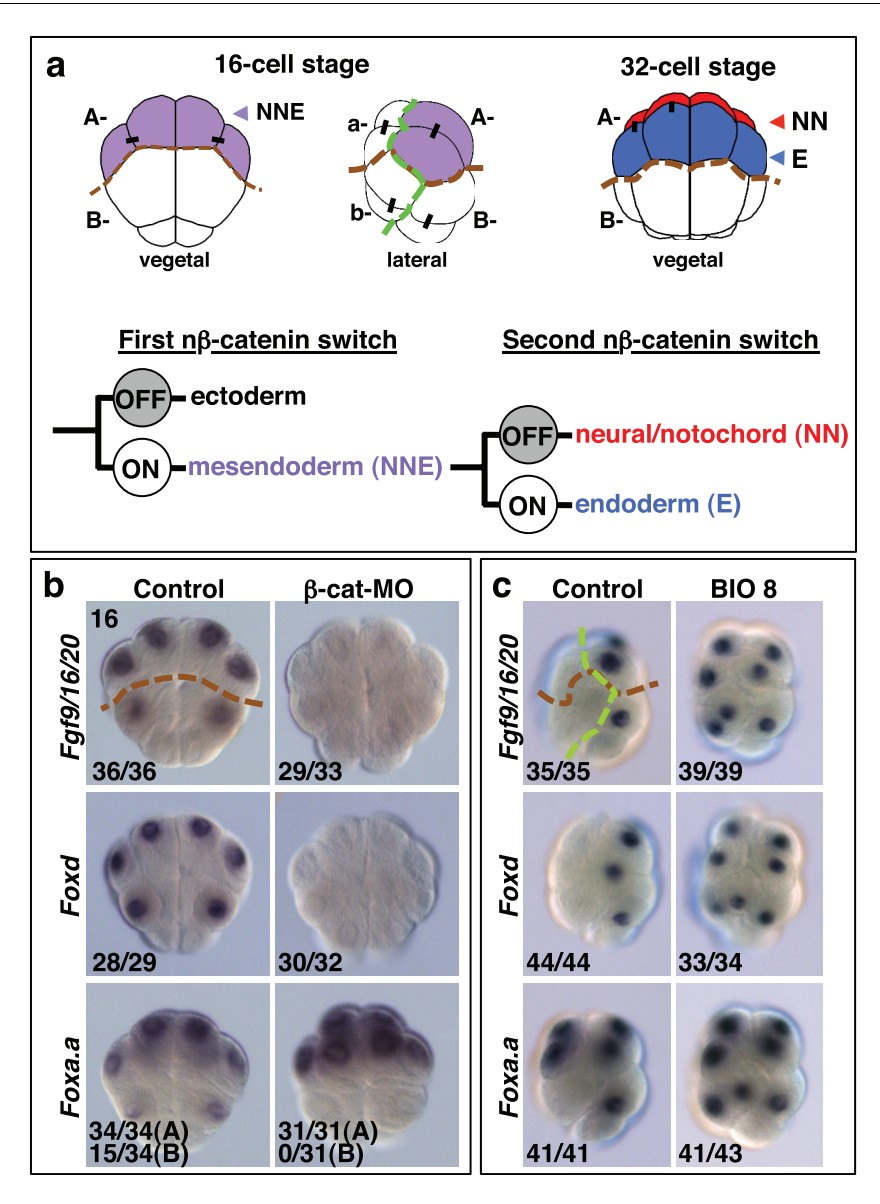

**Figure 1.** *Foxa.a, Foxd* and *Fgf9/16/20* are candidate NNE lineage specification factors. (a) Schematic drawings of embryos at the 16- and 32-cell stages. In this and all subsequent figures, where shown, a green dashed line separates the animal (ectoderm) from the vegetal (mesendoderm) hemispheres and a brown dashed line separates A- (A4.1) and a- (a4.2) lineages from B- (B4.1) and b- (b4.2) lineages. Different embryonic founder lineages are indicated on the drawings. NN and E cells are indicated in red and blue, respectively. Below the embryo drawings is a schematic representation of the two rounds of nβ-catenin-driven binary fate decisions that segregate firstly the mesendoderm lineages from the ectoderm lineages at the 16-cell stage and secondly segregate the mesoderm (NN) lineages from the endoderm (E) lineages at the 32-cell stage (*Hudson et al., 2013*). (b, c) Embryos analysed at the 16-cell stage for the marker indicated to the left of the panels following the treatment indicated above the panels [(b) vegetal pole view; (c) lateral view, vegetal pole to the right]. The numbers on the bottom-left corner of each panel indicate the proportion of embryos that the panel represents. The posterior most cells (at the bottom of the panels) are transcriptionally quiescent cells that will generate the germ line (*Shirae-Kurabayashi et al., 2011*). For *Foxa.a* expression in (b) control embryos showed expression in all four A-line (NNE) cells in 34/34 embryos, and in B-line cells, in 15/34 embryos, as indicated, whereas β-catenin-MO (β-cat-MO) injected embryos showed expression in NNE cells (31/31), but not B-line (0/31). Expression of *Foxa.a* in the four a-line precursors (not visible in the image) was not affected by β-catenin-MO injection.

embryo along the A-V axis at the 8- and 32-cell stages can be considered as the earliest steps of germ layer segregation.

β-catenin is a transcriptional co-activator which acts in a complex with TCF DNA-binding proteins to mediate the canonical Wnt signalling pathway (*Valenta et al., 2012*). The β-catenin/TCF complex promotes endoderm or mesendoderm in a wide range of organisms, and this process is therefore likely to represent an ancestral mechanism (*Darras et al., 2011*; *Henry et al., 2008*; *Hudson et al., 2013*; *Imai et al., 2000*; *Logan et al., 1999*; *McCauley et al., 2015*; *Miyawaki et al., 2003*; *Momose and Houliston, 2007*; *Wikramanayake et al., 1998*, *2003*). We have previously shown that the earliest steps of germ layer segregation in ascidian embryos are mediated by two rounds of nuclear(n)-β−catenin-dependent binary fate decisions. The first nβ-catenin-driven binary fate decision takes place at the 8- to 16-cell stage. During this process, the β-catenin/TCF complex is differentially activated between mesendoderm and ectoderm progenitors, resulting in segregation of these lineages (*Figure 1a*) (*Hudson et al., 2013*; *Oda-Ishii et al., 2016*; *Rothbächer et al., 2007*). The second step takes place at the 32-cell stage and controls the segregation of NNE mesendoderm cells into endoderm (E cell) and notochord/neural (NN cell) lineages (*Hudson et al., 2013*). During this step, the β-catenin/TCF complex is again differentially activated between E and NN cells (*Figure 1a*). Therefore, cells in which nβ-catenin remains active during the two steps (ON + ON) are specified as endoderm lineage, cells in which nβ-catenin remains inactive during the two steps (OFF + OFF) are specified as ectoderm lineage and cells in which nβ-catenin is active during the first step but inactive during the second step (ON + OFF) are specified as notochord-neural lineage (*Hudson et al., 2013*). These two rounds of nβ-catenin-driven switches result in transcriptional activation of the lineage specifiers, *Zic-related.b* (*Zic-r.b*, formally *ZicL*) and *Lhx3/4* (formally *Lhx3*), in NN and E cells, respectively (*Imai et al., 2002c*; *Satou et al., 2001*). One of the key features of these reiterative nβ-catenin-driven binary fate decisions is that the same asymmetric cue (nβ-catenin) is interpreted differently during each step (*Bertrand and Hobert, 2010*). Thus, in the NNE lineage, it is likely that the transient regulatory state induced by the first nβ-catenin input in NNE cells confers a distinct transcriptional response to the second nβ-catenin input on E cells.

In this study, we characterise the NNE lineage specification factors, which are induced by the first nβ-catenin input and address how these mesendoderm factors feed into the gene regulatory network of the NN and E lineages.

## Results

### *Foxa.a*, *Foxd* and *Fgf9/16/20* are nβ-catenin transcriptional targets in NNE cells

Following the first nβ-catenin activation at the 16-cell stage, *Foxa.a*, *Foxd*, *Fgf9/16/20*, *cadherinII* and βCD1 (β-catenin downstream gene 1) are induced in the NNE cells, with at least *Foxd* and *Fgf9/16/20* being direct targets of the β-catenin/Tcf7 complex (*Imai, 2003*; *Imai et al., 2002a*, *2002b*, *2002c*; *Kumano et al., 2006*; *Oda-Ishii et al., 2016*; *Rothbächer et al., 2007*; *Satou et al., 2001*). Consistent with a recent study (*Oda-Ishii et al., 2016*), we confirmed that in β-catenin–inhibited (β-catenin-MO injected) embryos analysed at the 16-cell stage, *Foxd* and *Fgf/9/16/20* expression was lost (*Figure 1b*). In addition to the mesendoderm lineages, *Foxa.a* is also expressed in the a-line anterior ectoderm lineages in a nβ-catenin-independent fashion (*Figure 1b,c*) (*Lamy et al., 2006*). In β-catenin–inhibited embryos, *Foxa.a* expression persisted in NNE and a-lineage cells, probably due to transformation of vegetal cells into animal cells that has been reported previously (*Figure 1b*) (*Imai et al., 2000*; *Oda-Ishii et al., 2016*). Conversely, ectopic stabilisation of nβ-catenin resulted in activation of all three genes in ectoderm lineages at the 16-cell stage (*Figure 1c*). This was achieved by treating embryos with BIO, a chemical inhibitor of the upstream inhibitory regulator of β-catenin, GSK-3, from the eight-cell stage (*Meijer et al., 2003*). Thus, our results confirm that *Foxd*, *Foxa.a* and *Fgf9/16/20* are transcriptional targets of nβ-catenin in vegetal cells, although *Foxa.a* also has a nβ-catenin-independent expression in a-line animal cells.

## *Foxa.a, Foxd* and *Fgf9/16/20*-signals are required for the correct initiation of both NN and E gene expression

It is likely that these gene products, activated by the first nβ-catenin signal in NNE cells, act together with the second differential nβ-catenin signal to activate the distinct gene regulatory networks between NN and E cells. Consistent with this idea, *Foxa.a* has been shown to be required for both NN lineage and endoderm gene expression (*Imai et al., 2006*), with *Foxd* specifically required for NN lineage, but not endoderm fates, and *Fgf9/16/20* contributing to notochord induction from the NN lineage (*Imai et al., 2002a*, *2002b*; *Yasuo and Hudson, 2007*). However, we found that inhibiting any one of these factors prevented the correct initiation of gene expression in both NN (*Zic-r.b*)

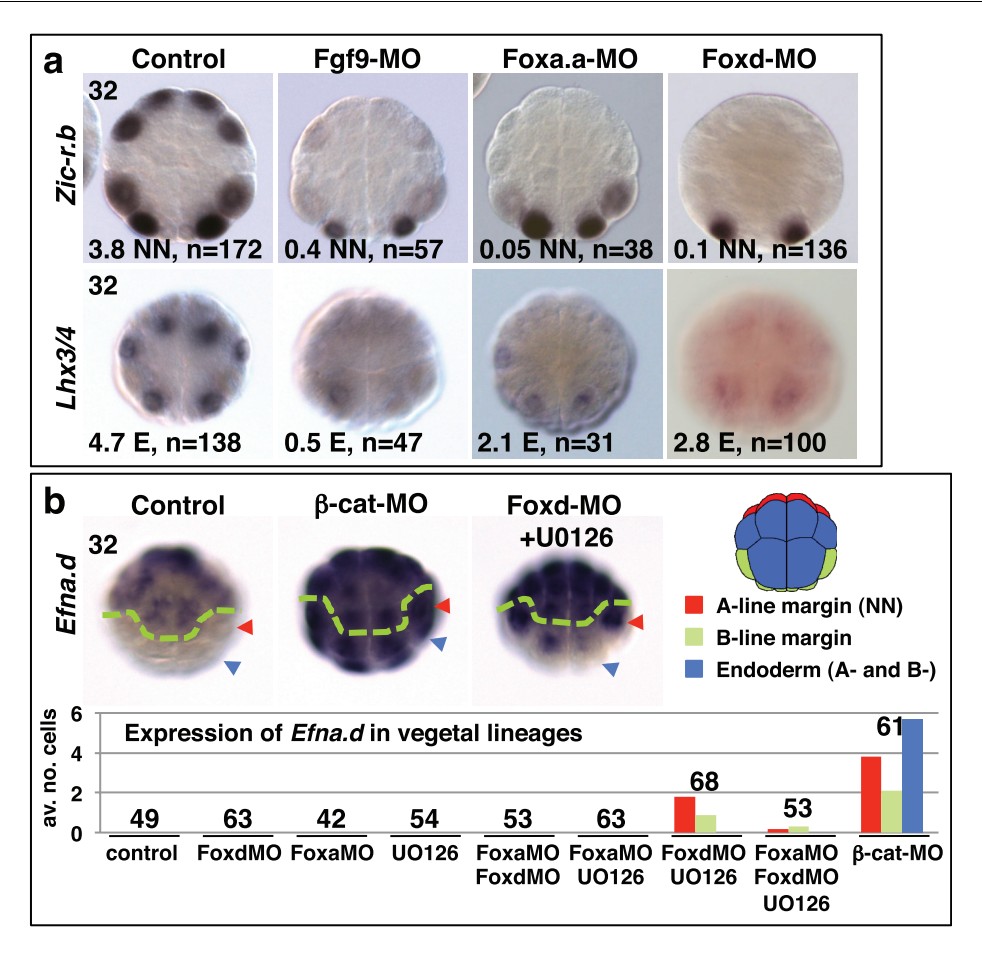

**Figure 2.** *Foxa.a, Foxd and Fgf9/16/20* are required for initiation of NN and E gene expression. (**a**) Embryos analysed at the 32-cell stage. The marker analysed is indicated on the left of the panels and the treatment indicated above the panels. The average number of NN (*Zic-r.b*) or E (*Lhx3/4*) cells expressing detectable levels of each gene is indicated. This remaining expression was generally weaker than control level expression. 'n=' represents the number of embryos analysed. (**b**) Expression of *Efna.d* under the conditions indicated. Embryos are shown in notochord-side view, animal pole up. The graph shows the average number of cells expressing *Efna.d* in different vegetal lineages at the 32-cell stage, as indicated by the key. All embryos showed ectoderm expression. The number of embryos analysed is indicated above the bars on the graph.

The following figure supplements are available for figure 2:

**Figure supplement 1.** *Foxa.a, Foxd and Fgf9/16/20* are required for initiation of NN and E gene expression.

**Figure supplement 2.** Endoderm formation under various conditions.

and E (*Lhx3/4*) lineages (*Figure 2a*, *Table 1*). We inhibited these factors using Morpholino anti-sense oligonucleotides (*Foxd-MO*, *Foxa.a-MO*, *Fgf9-MO*) and analysed *Zic-r.b* and *Lhx3/4* expression at the 32-cell stage, when NN and E cell lineages become segregated. FGF signals are frequently mediated by the MEK/ERK signalling pathway, leading to transcriptional activation via ETS family transcription factors, as is the case in ascidian embryos (*Bertrand et al., 2003*; *Kim and Nishida, 2001*; *Miya and Nishida, 2003*; *Yasuo and Hudson, 2007*). We confirmed that Fgf9/16/20 is responsible for the broad activation of ERK at the 32-cell stage in most vegetal lineages, including NN and E lineages, as well as two neural lineages in the ectoderm (*Figure 2—figure supplement 1f*). Treatment of embryos from the 16-cell stage with the MEK inhibitor U0126, also inhibits this ERK1/2 activation (*Kim and Nishida, 2001*; *Picco et al., 2007*). Inhibition of Fgf9/16/20, MEK or ETS1/2 (ETS1/2-MO) gave similar results, although inhibition of ETS1/2 gave only a weak down-regulation of *Zic-r.b* expression at the 32-cell stage, perhaps indicating the involvement of additional transcription factors that are also known to mediate FGF signals in *Ciona* embryos (*Figure 2a*; *Table 1*) (*Bertrand et al., 2003*; *Gainous et al., 2015*). Maintenance of *Foxa.a*, *Foxd* and *Fgf9/16/20* expression at the 32-cell stage is independent of each other (*Figure 2—figure supplement 1a*), as was shown previously for *Foxd* and *Fgf9/16/20* in *Ciona savigni* embryos (*Imai et al., 2002a*).

In FGF-inhibited embryos, *Zic-r.b* expression recovered at the 64-cell stage (*Figure 2—figure supplement 1a*) (*Imai et al., 2006*; *Kumano et al., 2006*). *Zic-r.b* expression at the 32- and 64-cell stages can be mediated by separate enhancer elements (*Anno et al., 2006*). In addition, in the NN-cell lineage, FGF-signalling is required for notochord fate, but has to be attenuated for neural fate (Minokawa et al, 2001; *Picco et al., 2007*; Yasuo and Hudson, 2007). Thus, an FGF-independent expression of *Zic-r.b* at the 64-cell stage, at least in neural fated cells, is not unexpected. In *Foxa.a*- and *Foxd*- inhibited embryos, *Zic-r.b* continues to be repressed in the NN-cell lineages at the 64-cell stage (*Figure 2—figure supplement 1a*) and later (*Imai et al., 2006*), consistent with a requirement for *Foxa.a* and *Foxd* for both NN cell lineage-derived structures, the notochord and caudal central nervous system (CNS) (*Imai et al., 2002b*, *2006*).

Endoderm gene expression was continuously reduced up to at least the early gastrula stage, following inhibition of any one of the NNE factors (*Figure 2—figure supplement 1a–e*). However, using alkaline phosphatase activity as an indicator of endoderm formation (*Whittaker, 1977*), a complete loss of endoderm at larval stages was observed only in *Foxa.a*-inhibited embryos, consistent with previous studies (*Figure 2—figure supplement 2*) (*Imai et al., 2002b*, *2006*). In *Foxd* and FGF-signal inhibited embryos, a large domain of alkaline phosphatase activity could be detected, suggesting that endoderm fate recovers in these embryos (*Figure 2—figure supplement 2*). Simultaneous repression of *Foxd* and FGF-signalling, however, resulted in both a stronger repression of early endoderm gene expression as well as an almost complete absence of alkaline phosphatase activity at larval stages (*Figure 2—figure supplement 1–2*). Thus, for eventual endoderm formation, the embryo is able to compensate for loss of either Foxd or FGF-signals but is not able to compensate for loss of both.

As well as promoting vegetal 'mesendoderm' fates, nβ-catenin also represses the ectoderm gene programme in vegetal cells (*Hudson et al., 2013*; *Imai et al., 2000*; *Oda-Ishii et al., 2016*; *Rothbächer et al., 2007*). In β-catenin knock-down embryos, ectopic expression of the early ectoderm gene *Efna.d* (formally *ephrin-Ad*) is observed in both NN and E cells at the 32-cell stage (*Figure 2b*) (*Hudson et al., 2013*) as well as in NNE cells at the 16-cell stage (*Oda-Ishii et al., 2016*). Double inhibition of *Foxd* and FGF-signals also resulted in ectopic expression of *Efna.d* in NN cells, but never in E cells (*Figure 2b*). Thus, NNE factors repress the ectoderm genetic programme in NN

**Table 1.** Expression of *Zic-r.b* in NN cells and *Lhx3/4* in E cells of 32-cell stage embryos, following inhibition of Fgf-signalling components.

|  | Control | U0126 | ETS1/2-MO |
|---|---|---|---|
| *Zic-r.b* NN cell | 4.0 cells (n = 163) | 0.75 cells (n = 58) | 3.4 cells* (n = 74) |
| *Lhx3/4* E cell | 3.9 cells (n = 153) | 2.1** cell (n = 45) | 0.7 cells (n = 92) |

* 44/74 embryos exhibited weaker levels of *Zic-r.b* expression compared to controls.

**Remaining expression was weaker than control levels of expression.

cells. A lack of derepression in E cells is probably due to the presence of nβ-catenin in E cells at the 32-cell stage (*Hudson et al., 2013*), suggesting that nβ-catenin can repress the ectoderm genetic programme both via and independently of the NNE factors.

## Combinatorial activity of *Foxa.a*, *Foxd and Fgf9/16/20* induces a mesendoderm state

Our data so far show that *Foxa.a, Foxd and Fgf9/16/20*-ERK1/2 are individually required for the correct initiation of the genetic programmes of both NN and E cell lineages. Indeed, co-expression of these three factors takes place only in mesendoderm lineages, the NNE and B-line mesendoderm lineages of the 16- and 32-cell stage embryo (*Figure 1; Figure 2—figure supplement 1a*). At the 32-cell stage, the E cells continue to express these three genes, while NN cells express only *Foxa.a* and *Fgf9/16/20*. However, we have previously shown that *Foxd* transcripts preferentially segregate into the NN cells during the NN-E cell division, before they rapidly disappear (*Hudson et al., 2013*). Thus, NN cells also contain *Foxd* transcripts early in their cell cycle. We conclude that *Foxa.a, Foxd* and *Fgf9/16/20* are co-expressed only in mesendoderm lineages. We next addressed whether these three factors were sufficient to induce a mesendoderm regulatory state.

As well as in vegetal cells, *Foxa.a* is also expressed in a-line anterior animal cells and ERK1/2 is activated in one pair of a-line cells (the a6.5 pair) at the 32-cell stage (*Figure 3a*) (*Hudson et al., 2003*; *Lamy et al., 2006*). Thus, a6.5 cells possess two of the three mesendoderm lineage specifiers and yet they do not adopt an NNE-like lineage. Consistent with the notion that coexpression of *Foxa.a, Foxd and Fgf9/16/20* represents a NNE regulatory state, reintroduction of the remaining factor, *Foxd*, by mRNA injection, was able to convert a-line cells to a mesendoderm state (*Figure 3*). Injection of *Foxd* mRNA resulted in ectopic expression of *Zic-r.b* in a-line cells at the 32-cell stage (*Figure 3b*) (*Imai et al., 2002c*). Similarly, expression of *Bra,* a marker of notochord precursors, was induced at the 64-cell stage (*Figure 3b*). The broad ectopic expression of *Zic-r.b* and *Bra* in the a- -lineage was clearly not restricted to the a6.5 cells. The most likely reason for this was that *Foxd* mRNA injection led to weak activation of *Fgf9/16/20* and strong inhibition of *Efna.d* expression in the ectoderm cells (*Figure 3—figure supplement 1*). Efna.d is a known antagonist of FGF-signals in *Ciona* and its inhibition results in widespread activation of ERK1/2 in ectoderm lineages (*Ohta and Satou, 2013*; *Picco et al., 2007*; *Shi and Levine, 2008*). Consistent with this, when *Foxd* mRNA-injected embryos were treated with U0126, the ectopic expression of *Zic-r.b* was reduced and *Bra* expression was completely suppressed, mimicking the effect of MEK inhibition on endogenous *Zic-r. b* and *Bra* gene expression (*Figure 3b*). Thus, injection of *Foxd* mRNA is sufficient to convert *Foxa.a/* ERK1/2-positive a-line cells into mesoderm.

Injection of *Foxd* mRNA, however, was not sufficient to induce endoderm gene expression in ectoderm cells (*Figure 3c*). This was expected since the two-step nβ-catenin binary fate decision model predicts that segregation of the endoderm lineage from the NNE lineage requires a second round of nβ-catenin activation (*Figure 1a*) (*Hudson et al., 2013*). Accordingly, when *Foxd*-mRNA injected embryos were treated from the late 16-cell stage with BIO in order to mimic the second input of nβ-catenin activation, both *Zic-r.b* and *Bra* were repressed and *Lhx3/4* ectopically activated in the a-line ectoderm cells (*Figure 3b–c*). Thus, the NNE-like state induced in animal cells by *Foxd* mRNA injection behaves in the same way as NNE state of unmanipulated embryos.

Taken together, these experiments provide strong evidence that the combinatorial activity of *Foxa.a, Foxd and Fgf9/16/20*-ERK1/2 represents a NNE mesendoderm regulatory state downstream of the first round of nβ-catenin input. To further test this model, we addressed whether co-expression of *Foxa.a, Foxd* and *Fgf9/16/20* was sufficient to rescue mesoderm in β-catenin-knockdown embryos. β-catenin-MO injected embryos would express only *Foxa.a* among the three genes (*Figure 1b*). We have shown that injection of *Foxd* mRNA results in induction of low levels of *Fgf9/16/20* expression, together with a strong suppression of *Efna.d* expression (*Figure 3—figure supplement 1*). Thus, injection of *Foxd* mRNA should be sufficient to recapitulate a *Foxd/Foxa.a/Fgf9/16/ 20* overlap. Consistent with this, injection of *Foxd* mRNA was able to rescue expression of NN-lineage genes (*Zic-r.b* and *Bra*) in β-catenin-MO embryos and, as expected, this recovery depended on an intact FGF-signalling pathway (*Figure 4*). We conclude that co-expression of *Foxd+Foxa.a+Fgf9/ 16/20* is sufficient to induce a mesendoderm regulatory state, which can then be further programmed into mesoderm or endoderm lineage by manipulation of nβ-catenin activity.

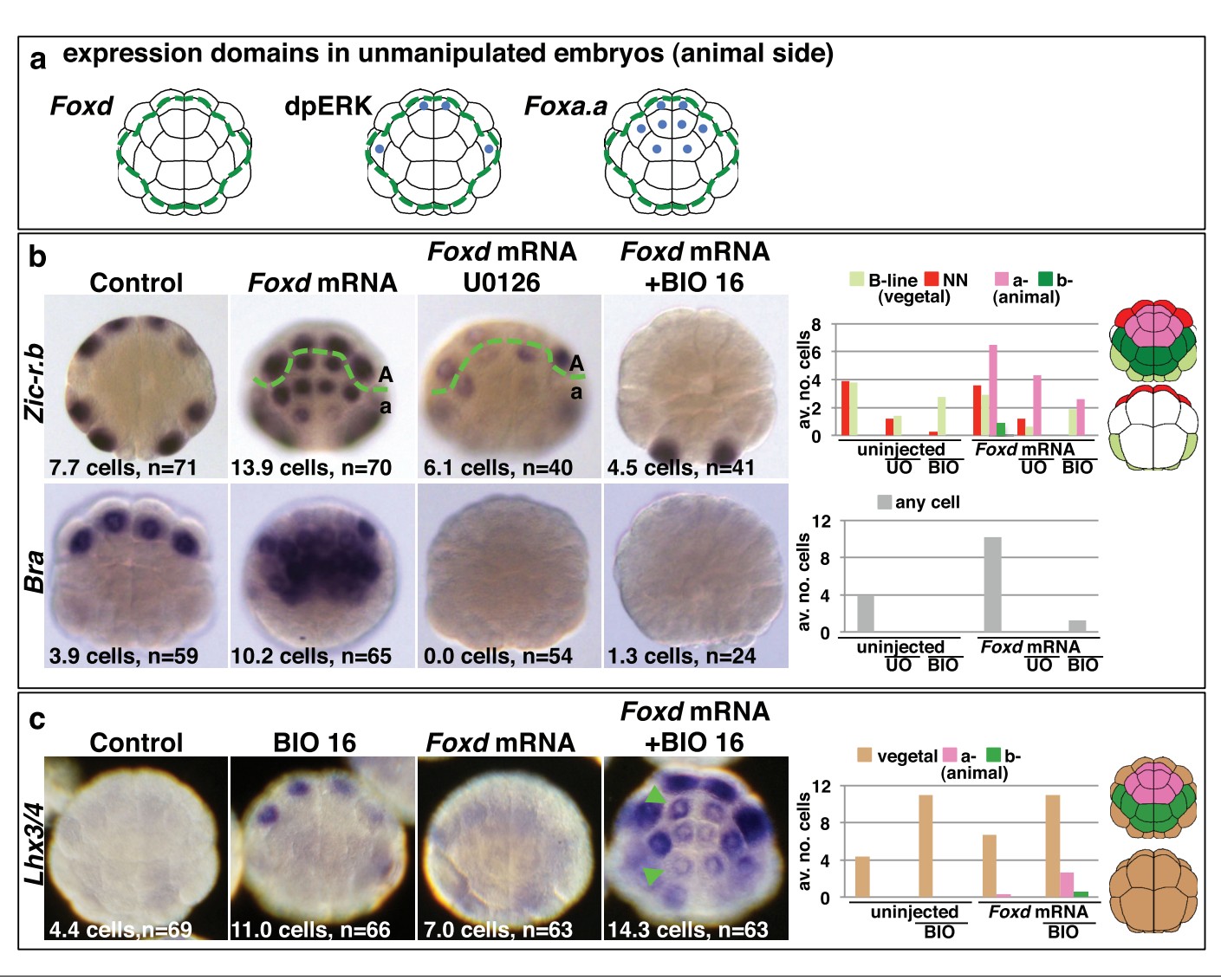

**Figure 3.** Creation of an ectopic *Foxa.a, Foxd, FGF*-signal overlap leads to ectopic mesendoderm formation. (**a**) Schematics show endogenous ectodermal expression of *Foxa.a, Foxd* (no expression) and activation of ERK (dpERK), indicated by blue dots. (**b–c**) Treatment is indicated above the panels and marker analysed to the left. All embryos in animal pole view except control *Bra* (vegetal pole view). Numbers show the total average number of cells per embryo expressing each marker. n = total number of embryos analysed. The graphs show the average number of cells expressing each maker in the lineages indicated on the keys, following the treatments indicated of the x-axis. No *Zic-r.b* expression was detected in endoderm lineages. In (**c**), the green arrowheads highlight the eight a-lineage cells. For (**b**), representative panels of uninjected/UO-treated and uninjected/BIO-treated embryos are not shown. The numbers of these experiments are: for *Zic-r.b*- U0126 alone n = 40 (average number of cells 2.6), BIO-16 alone n = 40 (average number of cells 2.9) and for *Bra*- U0126 alone, n = 39 (average number of cells 0.0); BIO-16 alone n = 31 (average number of cells 0.0).

The following figure supplement is available for figure 3:

**Figure supplement 1.** *Foxd* mRNA injection leads to repression of *Efna.d* and upregulation of *Fgf9/16/20* in ectodermal cells at the 16-cell stage.

### *Foxa.a, Foxd* and *Fgf9/16/20* act synergistically to reprogramme developing ectoderm cells to a mesendoderm state

We next addressed whether ectopic expression of *Foxd, Foxa.a* and *Fgf9/16/20* was able to reprogramme developing ectoderm to a mesendoderm state (**Figure 5**, **Figure 5—figure supplement 1**). The upstream regulatory sequences of the *Fucosyltransferase-like* (FT) gene become active in ectoderm cells from the 64-cell stage, when the ectoderm genetic programme is already underway

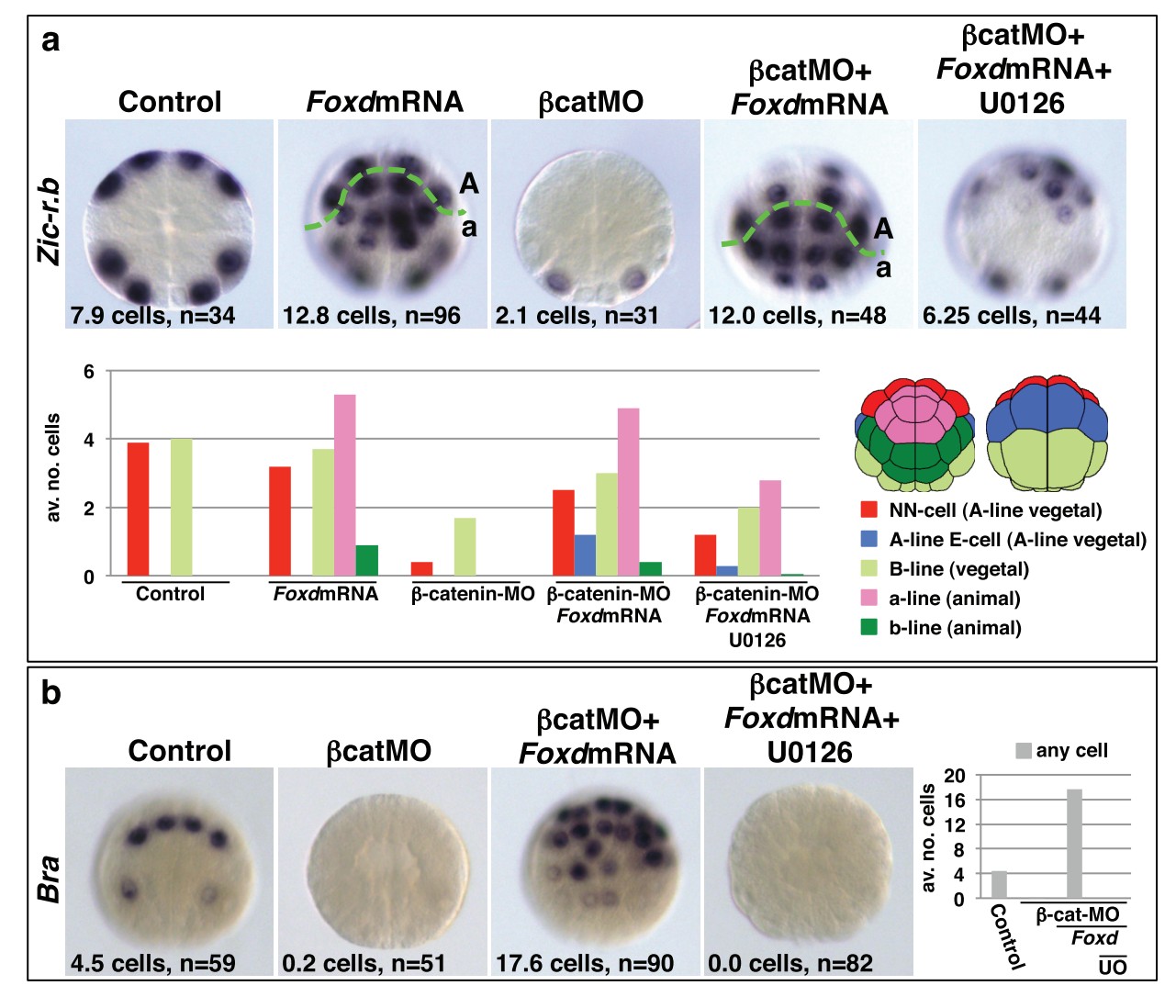

**Figure 4.** *Foxd* mRNA injection rescues mesoderm in β-catenin-MO injected embryos. (a–b) Treatment is indicated above the panels and marker analysed to the left of the panels. The total average number of cells per embryo is indicated, 'n=' indicates the total number of embryos analysed for each treatment. The graphs show the average number of cells expressing each marker in the lineages indicated by the keys, following the treatments indicated.

(*Figure 5—figure supplement 1*) and when these cells no longer express *Foxa.a*, *Foxd* or *Fgf9/16/20* (*Imai et al., 2004*; *Pasini et al., 2012*). Using FT promoter-driven constructs (*pFT>Foxa.a, pFT>Foxd* and *pFT>Fgf9/16/20*), we expressed *Foxa.a*, *Foxd* and *Fgf9/16/20* in different combinations in ectoderm lineages. To simplify the analysis and to rule out the possibility that signals from the vegetal cells may influence the experimental outcome, animal hemispheres of electroporated embryos were isolated by micro-dissection at the eight-cell stage. Isolated explants were cultured until the neurula stage when they were assayed for *Bra* expression (*Figure 5a–b*). *Bra* was chosen for this assay for its mesoderm (notochord)-specific expression. We observed a clear combinatorial effect between *Foxa.a*, *Foxd* and *Fgf9/16/20* on the reprogramming of ectoderm to mesoderm, with strong induction of *Bra* seen only when all three constructs were co-electroporated (*Figure 5b*). This reprogramming was accompanied by a strong downregulation of ectoderm gene expression and ectopic expression of *Zic-r.b* in the ectoderm cells of whole embryos (*Figure 5—figure supplement 1*). Furthermore, ectoderm explants could be reprogrammed to adopt an endoderm state (*Figure 5c*). To achieve this, ectoderm explants

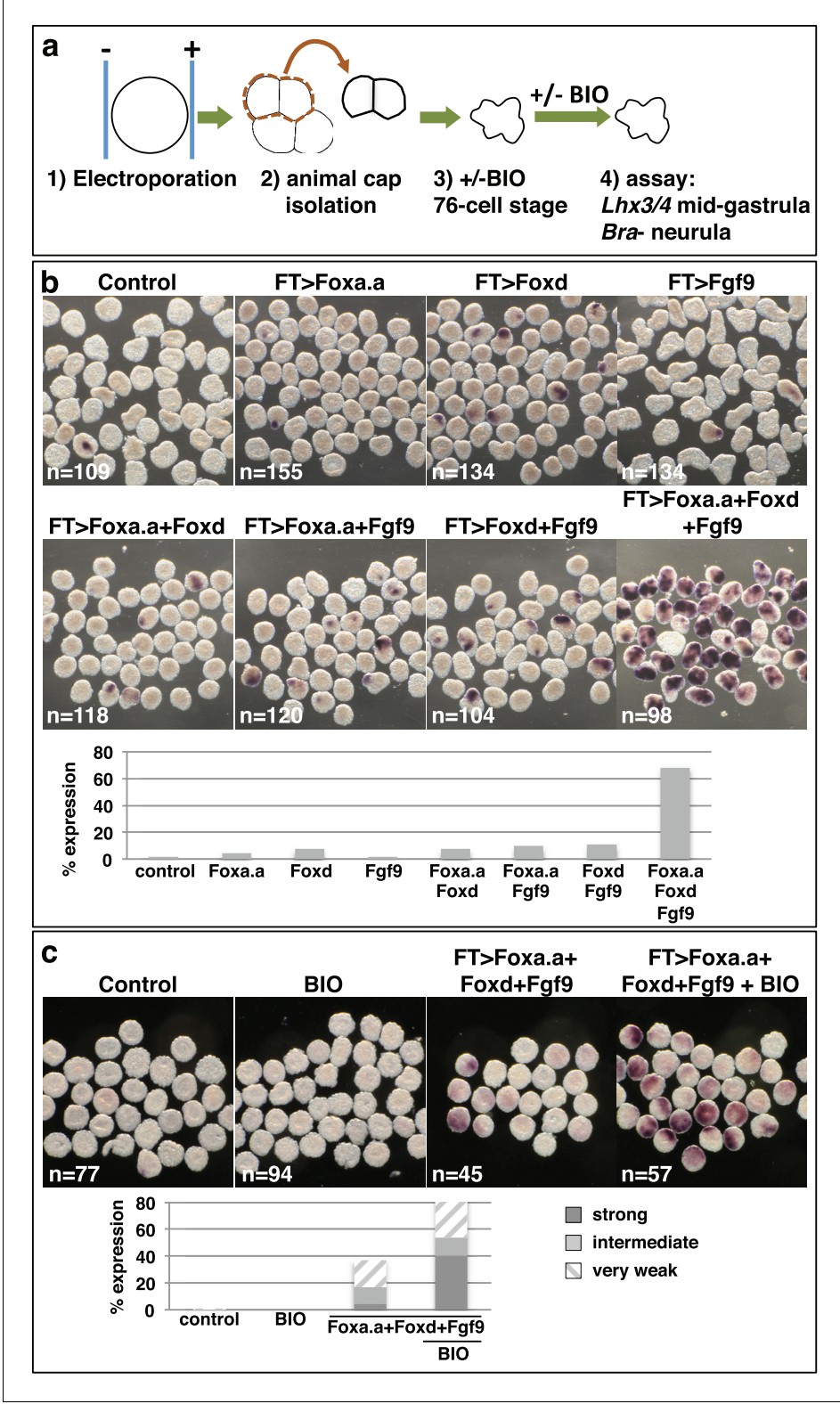

**Figure 5.** Reprogramming the ectoderm lineage to mesendoderm. (a) Experimental scheme. Embryos were electroporated and the ectoderm lineage (animal cap) isolated at the eight-cell stage. Ectodermal explants were cultured until the mid-gastrula stage for *Lhx3/4* expression or until the neurula stage for *Bra* expression. Optionally, explants were treated with BIO, when control sibling embryos reached the 76-cell stage, for approximately 1 hr prior to fixation (*Lhx3/4* only). (b–c) Expression of *Bra* (b) and *Lhx3/4* (c) in isolated ectodermal

*Figure 5 continued*

explants, following the treatments indicated above the panels. 'n=' represents the number of explants analysed. Graphs shows the percentage of explants with any level of *Bra* expression or level of *Lhx3/4* expression indicated by the key, under various conditions (*Foxa.a*= pFT>*Foxa.a*; *Foxd* = pFT>*Foxd*; *Fgf9*= pFT>*Fgf9/16/20*; control = unelectroporated).

The following figure supplements are available for figure 5:

**Figure supplement 1.** Reprogramming of ectoderm cells to mesendoderm fates.

**Figure supplement 2.** Confirmation that BIO-treatment of ectoderm explants at the 76-cell stage results in nuclear localisation of β-catenin.

from embryos electroporated with the triple combination (*pFT>Foxa.a, pFT>Foxd* and *pFT>Fgf9/16/20*) were treated with a pulse of BIO from the 76-cell stage to mimic the second round of nβ-catenin activation that normally drives the segregation of NN and E lineages (*Figure 5c*). The 76-cell stage was chosen, as at this stage ectopic expression driven by the *pFT* constructs is readily detectable (*Figure 5—figure supplement 1a*), the ectoderm programme is downregulated (*Figure 5—figure supplement 1b*) and ectopic *Zic-r.b* is not yet detected (*Figure 5—figure supplement 1c*). This stage thus represented the best approximation of the NNE state of normal embryos. We confirmed that BIO-treatment was able to induce nuclear translocation of β-catenin in isolated explants at the 76-cell stage (*Figure 5—figure supplement 2*). Endoderm induction was assayed by detection of *Lhx3/4* expression at the mid-gastrula stage, that is approximately 1 hr after the onset of BIO-treatment. Coupling these three factors with BIO-treatment resulted in strong induction of *Lhx3/4* expression. We conclude that the combinatorial activity of *Foxa.a, Foxd* and *Fgf9/16/20* is sufficient to reprogramme developing ectoderm cells to adopt a mesendoderm state.

## Discussion

In this study, we have identified *Foxa.a, Foxd and Fgf9/16/20* as the mesendoderm lineage specifiers of the NNE cell. Transcriptional activation of *Foxa.a, Foxd* and *Fgf9/16/20* is induced by the first nβ-catenin switch (*Figures 1a,6a*). Co-expression of these three factors is sufficient to reprogramme ectoderm cells to adopt a mesendoderm state. This ectopic mesendoderm state can be further converted into either mesoderm or endoderm by modulating nβ-catenin activation.

### A model for ascidian germ layer segregation

We propose the following model to summarise the initial stages of germ layer segregation in ascidian embryos (*Figure 6a*). At the 8- to 16-cell stage of development, nβ-catenin, activated specifically in vegetal cells by as yet unknown mechanisms, promotes *Foxa.a, Foxd* and *Fgf9/16/20* expression and represses ectoderm gene expression (*Hudson et al., 2013*; *Imai et al., 2000*; *Oda-Ishii et al., 2016*; *Rothbächer et al., 2007*). *Foxa.a, Foxd* and *Fgf9/16/20*, are co-expressed exclusively in mesendoderm lineages at the 16- to 32-cell stage of development (*Imai et al., 2002a, 2002b*; *Oda-Ishii et al., 2016*), where they are required, individually, for the correct initiation of both NN and E cell lineage gene expressions at the 32-cell stage (*Figure 2a*). The NNE factors are also required to repress ectoderm gene expression: co-inhibition of *Foxd* and *Fgf9/16/20* resulted in ectopic ectoderm gene expression in NN cells (*Figure 2b*) and *Foxd* overexpression alone was able to repress ectoderm gene expression (*Figure 3—figure supplement 1*). However, our data also suggest that nβ-catenin can repress ectoderm gene expression independently of these three factors (*Figure 2b*). Recently, it has been shown that this can take place via a physical interaction between β-catenin/Tcf7 and Gata.a, preventing this key regulator of ectoderm lineage from binding to its DNA target sites (*Oda-Ishii et al., 2016*; *Rothbächer et al., 2007*).

Following inhibition of *Foxa.a, Foxd* or *Fgf9/16/20*, both endoderm and mesoderm development is perturbed at later stages of development, although there is a redundancy between *Foxd* and FGF-signalling for the eventual recovery of endoderm (*Figure 2* and *Figure 2—figure supplement 1*, *Figure 2—figure supplement 2*) (*Imai et al., 2002a, 2002b, 2002c, 2006*; *Kumano et al., 2006*).

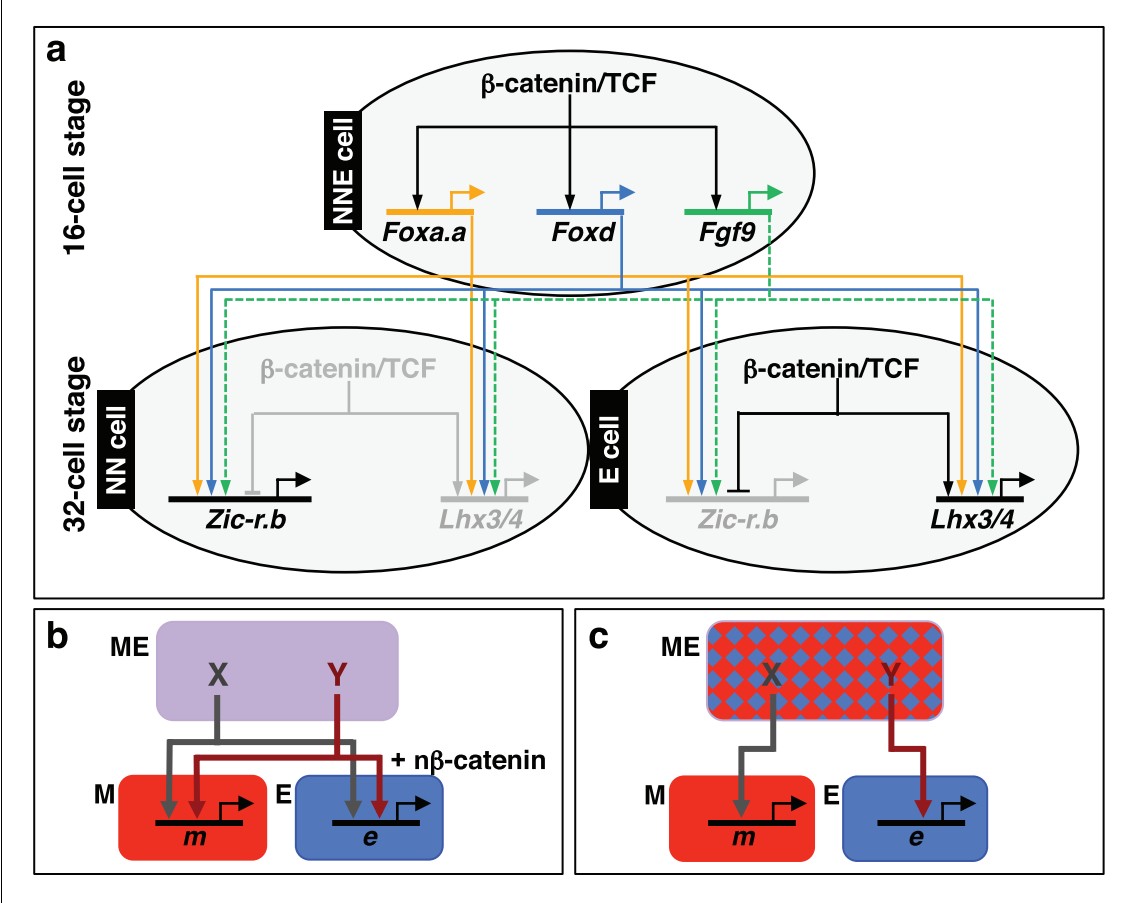

**Figure 6.** Gene regulatory model for segregation of NNE into NN and E lineages. (a) Each factor induced by nβ-catenin activation at the 16-cell stage feeds into both the NN and E lineage genes. The dashed line for Fgf9/16/20 represents a signalling molecule (most likely mediated, at least in part, by Ets1/2 transcription factor (*Table 1*). Differential gene expression between NN and E cells is mediated by the second nβ-catenin-driven switch. (b–c) Schematic regulatory architectures during mesendoderm segregation. ME = mesendoderm lineage; M = mesoderm lineage; E = endoderm lineage; e = endoderm gene; m = mesoderm gene; X, Y = genes expressed in mesendoderm cells. (b) Ascidian and nematode mesendoderm regulatory architecture. (c) 'Mixed-lineage' mesendoderm regulatory architecture.

It is likely that these factors play on-going roles during mesoderm and endoderm lineage progression. For example, ERK1/2 activity is detected in both notochord and endoderm until the early gastrula stage (*Nishida, 2003*; *Yasuo and Hudson, 2007*), *Fgf9/16/20* is required at the 64-cell stage for induction of notochord and repression of neural gene expression in the notochord lineage (*Imai et al., 2002a*; *Kim and Nishida, 2001*; *Minokawa et al., 2001*; *Yasuo and Hudson, 2007*) and *Foxa.a* is continuously expressed in notochord and endoderm, suggesting an on-going role for Foxa.a in both of these lineages (*Imai et al., 2004*).

Importantly, creating ectopic zones of co-expression of these three factors in distinct embryological settings, revealed their strong synergistic ability to induce a mesendoderm state, which can be further programmed to an NN or E-like lineage by modulation of nβ-catenin levels (*Figures 3–5* and *Figure 3—figure supplement 1*, *Figure 5—figure supplement 1*, *Figure 5—figure supplement 2*). We conclude that *Foxa.a, Foxd and Fgf9/16/20* are crucial for the mesendoderm ground state that canalises the daughter lineages to adopt either E or NN fates depending on the status of the second nβ-catenin input.

It is important to bear in mind that the germ layers are still not fully segregated at the 32-cell stage. While this manuscript has focused on the mesendoderm fates that arise from the NNE lineage, this lineage also produces neural tissue. NNE cells divide into E cells and NN cells. In addition to notochord, the NN cell generates the posterior part of the CNS, including the equivalent of the 'spinal cord' of

vertebrates (reviewed in [*Hudson, 2016*]). The binary cell fate decision between neural and notochord takes place at the 64-cell stage (*Minokawa et al., 2001*; *Picco et al., 2007*). The lateral neural progenitors that arise from the NN-cell lineage also produce a muscle cell during neural plate patterning, following another neuromesodermal binary fate decision (reviewed in [*Hudson and Yasuo, 2008*]). Bipotential neuromesoderm progenitors are not an ascidian novelty (*Henrique et al., 2015*; *Tzouanacou et al., 2009*). For example, in the zebrafish tailbud, bipotential neuromesodermal progenitor cells generate notochord and floorplate (ventral spinal cord) (*Row et al., 2016*) and in both human and mouse embryonic stem cells and zebrafish tailbud stem cells, bipotential neuromesodermal progenitors generate paraxial mesoderm and posterior neural tube (*Gouti et al., 2014*; *Martin and Kimelman, 2012*). Even in the classical mesendoderm model, that is the *C. elegans* EMS cell, the MS (mesoderm) lineage also gives rise to some neurons (*Sulston and Horvitz, 1977*; *Sulston et al., 1983*). The lateral E cells of *Ciona* are also not yet fate-restricted to endoderm fate. At the 64-cell stage of development, the lateral E cell divides into one endoderm and one trunk lateral cell (mesenchyme) precursor, following induction of trunk lateral cell fate (*Shi and Levine, 2008*). Thus, as in other species, ascidian germ layer segregation is an progressive process (*Tzouanacou et al., 2009*) and NNE specification should thus be considered as its first step.

## Regulatory architectures of mesendoderm

We have shown that, in ascidian embryos, individual mesendoderm lineage specifiers are required for the initiation of both mesoderm and endoderm GRNs (*Figure 6*). Furthermore, we have shown that the combinatorial activity of just three NNE factors is sufficient to reprogramme developing ectoderm cells to a mesendoderm state. The mesendoderm regulatory state in ascidian embryos is similar to the situation in the *C. elegans* EMS cell in which the MED1/2 GATA factors feed into both E (endoderm) and MS (mesoderm) lineage specification, such that MED1/2 directly activates both MS and E target genes (*Broitman-Maduro et al., 2005*; *Maduro et al., 2001*, *2015*; *McGhee, 2013*). Similarly, Foxa.a and Foxd can bind to the upstream sequences of both *Zic-r.b* (NN lineage) and *Lhx3/4* (E lineage), suggesting that this genetic interaction is direct (*Kubo et al., 2010*). While there is little doubt that mesendoderm transiently forms during embryogenesis of many animal models, and that both mesoderm and endoderm are induced by similar upstream regulators (β-catenin in invertebrates, β-catenin and Nodal in vertebrates), in most cases the transcriptional nature of the mesendoderm state does not appear to be similar to that of ascidians or nematodes. In particular, the existence of mesendoderm lineage specifiers (that is individual factors required for the initiation of both mesoderm and endoderm GRNs) have not been described in the majority of model organisms. For example, in sea urchins and anamniote vertebrates, mesendoderm has been described as a mixed regulatory state with simultaneous activation of mesoderm and endoderm GRNs, prior to the lineage segregation of these fates (*Peter and Davidson, 2010*; *Rodaway and Patient, 2001*). This type of 'mixed-lineage' regulatory architecture is also described in other systems and displays characteristics of multi-lineage priming, whereby the GRNs of two lineages are simultaneously activated prior to lineage segregation (*Figure 6c*) (*Graf and Enver, 2009*; *Nimmo et al., 2015*). If this were the scenario for the ascidian mesendoderm regulatory state, one would expect individual NNE factors to be required for, and be able to induce, only one or other of the two subsequent lineages (NN or E), but not both. A 'mixed-lineage' regulatory architecture is therefore not consistent with our data describing the NNE mesendoderm regulatory state (*Figure 6*).

These two regulatory architectures are, however, unlikely to be mutually exclusive. In sea urchin and sea stars, for example, genes interacting with both mesoderm and endoderm GRNs have been identified (http://sugp.caltech.edu/endomes/) (*Davidson et al., 2002*; *McCauley et al., 2015*). It cannot be ruled out that mesendoderm lineage specifiers, acting upstream of both endoderm and mesoderm GRNs, are more broadly utilised, but are simply difficult to uncover due to the sheer complexity of early embryos and their GRNs (*Ben-Tabou de-Leon and Davidson, 2009*; *Kiecker et al., 2016*; *Tremblay, 2010*). It is also possible that the regulatory architecture of nematode and ascidian mesendoderm resulted from an adaption to a lineage-based mode of development with small numbers of cells, perhaps enabling these rapidly developing embryos to bypass the need for cross-repression and prolonged stabilisations of the endoderm and mesoderm GRNs. In summary, it is not yet clear whether an obligate mesendoderm state (that is a state with mesendoderm lineage specifiers) is present in the majority of bilaterian developmental programmes, although this seems to be the case in nematode and ascidian embryos.

# Materials and methods

## Overexpression and knockdown tools

Morpholinos (MOs) were purchased from GeneTools (Philomath, Oregon) and have been reported previously: β-catenin-MO (*Hudson et al., 2013*); *Foxa.a-MO* and *Foxd-MO* (*Imai et al., 2006*); *Fgf9/16/20-MO* and *Fgf8/17/18-MO* (*Yasuo and Hudson, 2007*), ETS1/2-MO (*Bertrand et al., 2003*). *Foxa.a-MO* was injected at 0.85 mM and ETS1/2-MO at 0.75 mM. All other morpholinos were injected at 0.5 mM. U0126 was used at 2 µM and BIO (GSK-3 inhibitor IX) at 2.5 µM (both were purchased from Calbiochem (Merck, Darmstadt, Germany)). Since a full-length cDNA clone for *Ciona intestinalis Foxd* is not available in gene collection plates (*Gilchrist et al., 2015*; *Satou et al., 2002*), we synthesised the *Ciona savignyi Foxd* mRNA from pRN3-Cs-Foxd (*Imai et al., 2002b*) using mMESSEGEmMACHINE kit (Thermo Fisher Scientific, Waltham, MA). The *Ciona savigni Foxd* used in this study corresponds to Genbank accession number AB057738.1. *Foxd* mRNA was injected at 75ng/µl. In order to generate *pFT>Foxa.a*, *pFT>Foxd* and *pFT>Fgf9/16/20*, we first constructed Gateway (Invitrogen, a brand of Thermo Fisher Scientific) pENTR clones containing ORFs of these genes. ORFs of *Cs-Foxd, Foxa.a* and *Fgf9/16/20* were PCR-amplified using the following primer pairs and templates: Foxa.a-attB1 (aaaaag-caggctaccATGATGTTGTCGTCTCCACC) and Foxa.a-attB2 agaaagctgggtTTAGCTTGCTGGTACG-CAC) on cicl044j20 template; FGF9-attB1 (aaaaagcaggctaccATGTCTATGTTAACCAACATGTTAGG) and FGF9-attB2 (agaaagctgggtTCAGTAGAGTCGGCCAGAGTAC) on citb007k01; CsFoxd-attB1 (aaaaagcaggctaccATGACTGTGGACTCTTGTACAG) and CsFoxd-attB2 (agaaagctgggtCTAAATAAG TTTATACGGGAATGG) on pRN3-Cs-Foxd. The *Fucosyltransferase-like* driver has been reported previously (*Pasini et al., 2012*). The promoter region was PCR-amplified using the following pair of primers to generate a destination vector pSP1.72BSSPE-pFT::RfA-venus (*Roure et al., 2007*): pFT-attB3 (ggggacaagtttgtataataaagtaggctGGCATCATAACGTACAACCTG) and pFT-attB5 (ggggaccactttgta-tacaaaagtggggtTGCAGCGGTAGAGTTTACTATTATC). *pFT>Foxa.a*, *pFT>Foxd* and *pFT>Fgf9/16/20* were then generated by LR reaction between corresponding pENTR clones and pSP1.72BSSPE-pFT:: RfA-venus.

## Embryological experiments

Adult *Ciona intestinalis* were purchased from the Station Biologique de Roscoff (France). Blastomere names, lineage and the fate maps are previously described (*Conklin, 1905*; *Nishida, 1987*). Ascidian embryo culture and microinjection have been described (*Sardet et al., 2011*). All microinjections were carried out in unfertilised eggs. The electroporation protocol was based on *Christiaen et al., 2009*. Up to 60 µg of circular plasmid DNA was made up to 250 µl at 0.6 M mannitol. DNA/mannitol solution was mixed with 100 µl of eggs in artificial sea water supplemented with 0.5% BSA (to help prevent sticking). Electroporation was carried out at 50V for 16 ms using a BTX (Harvard Apparatus, Holliston, Massachusetts) ECM 830 and electroporated embryos transferred to agarose-coated dishes. For data shown in *Figure 5—figure supplement 1a*, 50 µg of *pFT>Foxd* was used. Otherwise, each FT construct was used at 20 µg to give a maximum total of 60 µg. *pFT>tdTomato* was used as a control electroporation at 60 µg. In all experiments, embryos that failed to develop were discarded and all other embryos scored. All data were pooled from at least two independent experiments (i.e., on different batches of embryos).

The experimental design of the BIO treatment of electroporated ectodermal explants, shown in *Figure 5c*, is as follows. Embryos were electroporated with three plasmids, *pFT>Foxa.a + pFT>Foxd + pFT>Fgf9/16/20*. Ectoderm explants were isolated at the eight-cell stage from control (unelectro-porated) and electroporated embryos. Each sample of explants was split into two groups and one group of each treated with BIO from the 76-cell stage. BIO treatment was continued until fixation, when sibling embryos reached the mid-gastrula stage. Explants remained in BIO for approximately 1 hr at 20°C.

## In situ hybridisation, gene naming, alkaline phosphatase staining and dpERK and β-catenin immunofluorescence

All gene markers used for in situ hybridisation have previously been described (*Hudson et al., 2013*; *Imai et al., 2004*) (http://ghost.zool.kyoto-u.ac.jp). According to recent nomenclature guidelines, we used *Zic-r.b* (previously called *ZicL*) to describe the five copies of *ZicL* gene named *Zic-r.b – Zic-r.f*,

*Lhx3/4* (previously *Lhx3*) and *Efna.d* (previously *ephrin-Ad*) (*Stolfi et al., 2015*). The in situ hybridisation and alkaline phosphatase staining protocols are previously described (*Hudson et al., 2013*). All single embryo panels, except those in *Figure 3c*, were mounted in 50–80% glycerol and photographed on an Olympus (Tokyo, Japan) BX51 using a Leica DFC310FX camera. All multi-embryo panels as well as single-embryo panels in *Figure 3c* were taken of embryos in PBT on a Leica (Leica microsystems, Vienna, Austria) Macroscope Z16 APO with a Canon (Tokyo, Japan) EOS 60D camera. The dpERK and β-catenin immunofluorescence protocols are described previously (*Haupaix et al., 2014*; *Hudson et al., 2013*). Immunostained embryos were mounted in Vectashield-DAPI (Vector laboratories, Burlingame, CA), analysed on a Leica SP5 confocal microscope and processed with Image J.

## Acknowledgements

We thank U Röthbacher and P Lemaire (FT promoter), H Nishida (β-catenin antibody), Y Satou (pRN3-Cs-Foxd and *Foxa.a-MO*) and N Satoh (Gene Collection Plates) for tools, Evelyn Houliston and David McClay for critical reading of the manuscript and helpful discussions and Jenifer Croce and David McClay for interesting discussions on sea urchin mesendoderm GRNs.

## Additional information

### Funding

| Funder | Grant reference number | Author |
| --- | --- | --- |
| Centre National de la Recherche Scientifique | | Hitoyoshi Yasuo |
| Université Pierre et Marie Curie | | Hitoyoshi Yasuo |
| Fondation ARC pour la Recherche sur le Cancer | #1144 | Hitoyoshi Yasuo |
| Agence Nationale de la Recherche | ANR-09-BLAN-0013-01 | Hitoyoshi Yasuo |
| Fondation ARC pour la Recherche sur le Cancer | PJA 20131200223 | Hitoyoshi Yasuo |

The funders had no role in study design, data collection and interpretation, or the decision to submit the work for publication.

### Author contributions

CH, HY, Conception and design, Acquisition of data, Analysis and interpretation of data, Drafting or revising the article, Contributed unpublished essential data or reagents; CS, Acquisition of data, Analysis and interpretation of data

### Author ORCIDs

Clare Hudson, http://orcid.org/0000-0003-1585-8328
Cathy Sirour, http://orcid.org/0000-0002-0114-8985
Hitoyoshi Yasuo, http://orcid.org/0000-0002-0817-3796

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
