## [Decision Letter]

[Editors’ note: this article was originally rejected after discussions between the reviewers, but the authors were invited to resubmit after an appeal against the decision.]

Thank you for submitting your work entitled "An essential transient mesendoderm state during germ layer segregation in ascidian embryos" for consideration by *eLife*. Your article has been reviewed by two peer reviewers, and the evaluation has been overseen by a Reviewing Editor and Janet Rossant as the Senior Editor. One of the two reviewers has agreed to reveal his identity: Brad Davidson.

Our decision has been reached after consultation between the reviewers. Based on these discussions and the individual reviews below, we regret to inform you that your work will not be considered further for publication in *eLife*.

Specifically, while the study is carefully carried out with nice experimental findings here, the reviewers felt that the overall hypothetical framework of a specific mesendodermal state was not well supported. Given this, the paper does not fundamentally change the way we think about the evolution of developmental regulatory networks and is perhaps more suited to a specialized developmental journal.

Reviewer #1:

Hudson et al. present a re-evaluation of established regulatory relationships between early regulators of endomesodermal fates in early ascidian embryo, using Ciona as a model and focusing on 32- and 64-cell stages.

The main point of the paper is to argue that FoxD, FoxAa and Fgf9/16/20 together form a mesendoderm-specific regulatory state activated downstream of β-catenin. One of the key novel experimental merits of the paper is to show that this cocktail of regulators is sufficient to reprogram ectoderm precursors into mesendodermal tissues, whether the mesodermal notochord or the endoderm, providing prolonged β-catenin activity for the latter.

Even though the experiments are well executed and reasonably interpreted, I have concerns about the whole premise of the paper:

It is difficult to call NNE precursors strictly mesendoderm, since the NN precursor will also produce posterior nerve chord. Rather it seems likely that these early progenitors are quite ascidian-specific even though they use conserved pan-bilaterian regulatory inputs such as the β-catenin/TCF regulatory axis.

For this cocktail of regulators to be specific to a mesendoderm state, one would expect them to be restricted to the pluripotent stage of the NNE progenitor. This seems to be the case for FoxD, however Fgf9/16/20 is later maintained in the NN progenitors and notochord, whereas FoxAa is later maintained in the endoderm. This suggests that – as shown to be the case for NN by Kobayashi et al., Genesis, 2013 – the NNE progenitors are primed by expressing early regulators of both the NN and the E fate. Thus, I am not convinced that the NNE "mesendoectoderm" progenitors depart significantly from a more classical model of multilineage priming of progenitors, as stated by the authors.

Finally, by the authors' admission, several of these regulatory linkages have been documented before, so it is not always clear what the novel insights really are.

Reviewer #2:

In this manuscript, the authors make substantial progress in delineating the transcriptional circuitry underlying mesendoderm lineage specification. Through an extensive and well-documented series of manipulations the authors demonstrate that three transcription factors are both necessary and sufficient for establishing the tunicate mesendodermal lineage. The authors then attempt to employ their results to help resolve questions about the nature and evolution of mesendodermal regulatory states in various clades. However, some of these interpretations are unclear and potentially inaccurate as discussed below.

My primary concern regards the overall framework the authors used to interpret their results. In the Abstract, the authors propose the mesendoderm may represent an "obligatory" or "essential" developmental state and indicate that their data supports this assertion. However, it is not clear that this is an accurate assessment.

According to this premise, the loss of the mesendoderm regulatory state should preclude the subsequent establishment of endoderm and mesodermal lineages. Yet, the authors’ data contradict this. Instead they found that loss of one or more key mesendodermal regulatory factors did not preclude the eventual establishment of endoderm or mesoderm. In particular, they showed that:

a) *ZicL* (a mesoderm lineage marker) expression was eventually restored after inhibition of FGF signaling (Figure 2);b) Endoderm differentiation (assayed by alkaline phosphatase activity) occurred despite inhibition of FGF or FoxD (Figure 2).

They also found that single "mesendoderm" factors were able to restore either mesoderm or endoderm even though were not able to restore the mesendoderm state, effectively bypassing this state and demonstrating that it is not essential or obligatory. In particular they showed that:

FoxD alone was able to restore expression of Mesoderm lineage genes after loss of the mesendoderm regulatory state through disruption of B-catenin (Figure 4).

There are at least three ways that the authors could address these concerns.

1) Clarify what they mean by an "essential" role for the mesendoderm and how their experimental results support this hypothesis.

2) Modify their hypothesis, possibly by suggesting that the three mesendodermal factors are locked into a recursive regulatory relationship. This caveat might explain why disruption of one factor only temporarily blocks the establishment of this regulatory state (Figure 2). It may also serve to explain why expression of a single factor is sufficient to re-establish the mesendodermal state/subsequent germ layer sub-specification (Figure 3). If they do propose this explanation, they should also evaluate whether it is supported by any data.

3) Discard their current hypotheses and focus on an alternative framework. For example, they could choose to focus their interpretation on different regulatory architectures that might underlie mesendoderm regulatory states in different embryos and some potential functional implications of these different architectures. This framework may allow a more productive comparison to other characterized endomesodermal circuits (nematode/sea urchins) as further discussed in the next point.

A secondary concern focuses on the authors' statement that a "distinct mesendodermal state" is not present in sea urchins (Introduction, first paragraph and Discussion, second paragraph). The sea urchin and sea star GRNs both include a conserved, transient mesendodermal state based on B-catenin dependent expression of wnt8 and Otx (Hinman, Yankura and McCauley 2009). The cited paper (Peter and Davidson, 2010) does state that:

"In a strict sense, early endomesoderm specification therefore involves the concomitant activation of two separate GRNs, rather than the activation of a common endomesoderm GRN which diverges into two daughter programs. The co-expression of regulatory genes to which we refer to as endomesoderm regulatory state is the result of these two GRNs."

However, this interpretation is not definitive and a good argument could be made that the endomesodermal state is established prior to the simultaneous activation of the two germ-layer specific programs, as laid out in the cited Hinman paper. The authors should clarify exactly what they think defines a distinct endodermal regulatory state and whether or not their data suggests a tunicate regulatory circuitry that is fundamentally distinct from that observed in sea urchins/sea stars.

---

## [Author Response]

[Editors’ note: the author responses to the first round of peer review follow.]

*Reviewer #1:*

Hudson et al. present a re-evaluation of established regulatory relationships between early regulators of endomesodermal fates in early ascidian embryo, using Ciona as a model and focusing on 32- and 64-cell stages.

[Please also refer to: “Finally, by the authors' admission, several of these regulatory linkages have been documented before, so it is not always clear what the novel insights really are.”]

This comment is in striking contrast to reviewer 2 who describes our work as ‘substantial progress’. Importantly, the point of our manuscript was to characterise the mesendoderm lineage specifiers and reveal exactly how these factors provide the ground state for the second round of β-catenin driven binary fate decisions that segregates the NN and E lineages and it was not possible to ascertain this from the data already published. A summary of the published data is summarised below and is described more carefully in the revised version of our manuscript.

In the literature at the time of our first submission it was reported that *Foxa.a, Foxd* and *Fgf9/16/20* are downstream of β-catenin and that *Foxd* is likely to be direct. Data showing that *Foxa.a* is a transcriptional target of β-catenin had never actually been documented, though it was previously mentioned in Imai et al., 2002b. The relationship between β-catenin and any of the three factors was also not reported at the 16-cell stage in relation to either loss or activation (BIO-treatment) of β-catenin. It was important to perform this analysis at the 16-cell stage, prior to fate segregation of the mesendoderm lineage into mesoderm and endoderm lineages at the 32-cell stage. While this data at the time was therefore novel (albeit expected), it has since shown by the group of Yutaka Satou that all three of these factors are downstream targets of β-catenin with at least *Fgf9/16/20* and *Foxd* being direct targets (Oda-Ishii et al., PLoS Genetics, 2016). This highly relevant manuscript is duly discussed in the revised version of our manuscript. We preferred to retain the, now mostly confirmatory, data in Figure 1 of our revised manuscript as it forms an important basis for our study.

To summarise the published data on the role of *Foxa.a, Foxd and Fgf9/16/20* in *Ciona*: Foxa.a was reported to be required for notochord and endoderm formation; Foxd for notochord but not endoderm and Fgf9/16/20 for notochord (partially) but not endoderm (Imai et al., 2002a, 2002b, Imai et al., 2006; Yasuo and Hudson, 2007). Similarly, in *Halocynthia* embryos Foxa.a, Foxd and FGF-signals were reported to be required for notochord formation (Kumano et al., 2006).

Important for our study was whether the expression of the NN and E lineage specifiers of the 32-cell stage embryo (*Zic-r.b* and *Lhx3/4* respectively) was affected by loss of any of the NNE factors in *Ciona*. However, only an examination of *Zic-r.b* expression at the 32-cell stage following Foxd-MO injection has been reported previously (Imai et al., 2002b). Thus, it was not known prior to our study whether Foxa.a, Foxd or FGF9/16/20 were involved in the correct initiation of NN and E cell genetic programmes. Thus, our data showing that all three factors are required individually for the initiation of gene expression in *both* NN *and* E cells is novel and also quite unexpected. Much of the supporting data, showing that endoderm gene expression is repressed at least until the early gastrula stage following Foxd or FGF-signal inhibition is also novel. The demonstration that Foxd and FGF-signals act in a partially redundant way during late endoderm specification is also novel, these factors were previously deemed dispensable for endoderm formation. Finally, the conversion of animal cells to an NNE-like state by co-expression of the three NNE factors, together with the demonstration that developing ectoderm cells can be reprogrammed to mesendoderm fate by co-expression of these three factors are novel.

In summary, while we agree that the regulatory relationships of these factors were partially known, the available data was not sufficient to answer our particular question and required further investigation. This should not be interpreted in any way as a criticism of the published manuscripts, which were addressing distinct questions to ours.

*The main point of the paper is to argue that FoxD, FoxAa and Fgf9/16/20 together form a mesendoderm-specific regulatory state activated downstream of β-catenin. One of the key novel experimental merits of the paper is to show that this cocktail of regulators is sufficient to reprogram ectoderm precursors into mesendodermal tissues, whether the mesodermal notochord or the endoderm, providing prolonged β-catenin activity for the latter.*

*Even though the experiments are well executed and reasonably interpreted, I have concerns about the whole premise of the paper:*

*It is difficult to call NNE precursors strictly mesendoderm, since the NN precursor will also produce posterior nerve chord. Rather it seems likely that these early progenitors are quite ascidian-specific even though they use conserved pan-bilaterian regulatory inputs such as the β-catenin/TCF regulatory axis.*

We agree with the reviewer that the NNE precursors are not strictly mesendodermal. However, we do not agree that the ascidian ‘neuro-mesendodermal’ cells are likely to be ascidian specific. Rather, a study of the literature reveals common origins for neural and mesoderm in many species and as in many other species, ascidian mesendoderm specification begins with β-catenin signalling. The ascidian NNE (Neural, Notochord, Endoderm) cell, specified by β-catenin signals, divides into a neuro-mesoderm (NN) and (mostly) endoderm (E) progenitor. The NN cell generates notochord (mesoderm) and the posterior part of the CNS, including the equivalent of the ‘spinal cord’ with its ventral cells that are potentially equivalent to the vertebrate floor plate. Formation of these tissues follows a binary cell fate choice between neural and notochord: not neural and epidermis. Similarly, in the zebrafish tailbud, bipotential progenitor cells generate notochord and floorplate (Row et al., 2016, Development); in both human and mouse embryonic stem cells and zebrafish tailbud stem cells, bipotential neuromesodermal progenitors generate paraxial mesoderm and posterior neural tube (Gouti et al., PLOS Biology, 2014; Martin and Kimelman, 2012; Dev. Cell.). Even in the classical mesendodermal model, that is the *C. elegans* EMS cell, the MS lineage also gives rise to neurons (Sulston and Horvitz, 1977, Developmental Biology; Sulston et al., 1983 Developmental Biology). Further, several mutations in mice result in fate changes between neural and mesoderm- *Fgfr1* mutant cells fail to pass through the primitive streak and form secondary neural tubes instead of mesoderm (Ciruna et al., 1997, Development); in Wnt-3a mutants, cells ingress through the primitive streak, but fail to form somites, instead forming secondary neural tubes (Yoshikawa et al., 1997, Developmental Biology); in Tbx6 mutants posterior somites are converted to extra neural tubes (Chapman and Papaioannou, 1998, Nature). While classically the mesoderm germ layer is called as such rather than neuromesoderm, we feel that it is highly likely, due to the widespread nature of the neuro-mesoderm bipotential choice, that some neural fates originates from the so-called the ‘mesoderm’ germ layer.

In response to this concern raised by reviewer 1, we have now discussed in more detail the process of germ layer segregation in ascidian embryos to make it clear that 1) it is a progressive process, 2) part of the neural tissue shares a common origin with mesoderm, and 3) these two points are similar to what is observed in other chordate embryos.

*For this cocktail of regulators to be specific to a mesendoderm state, one would expect them to be restricted to the pluripotent stage of the NNE progenitor. This seems to be the case for FoxD, however Fgf9/16/20 is later maintained in the NN progenitors and notochord, whereas FoxAa is later maintained in the endoderm. This suggests that – as shown to be the case for NN by Kobayashi et al., Genesis, 2013 – the NNE progenitors are primed by expressing early regulators of both the NN and the E fate. Thus, I am not convinced that the NNE "mesendoectoderm" progenitors depart significantly from a more classical model of multilineage priming of progenitors, as stated by the authors.*

We disagree. Firstly, we see absolutely no reason that a regulator can only be used once rather than for multiple rounds of cell lineage decisions (though this would be more convenient for developmental biologists). Importantly for our study it is the *combinatorial activity* of these three factors that specifies the NNE state and co-expression of all three factors takes place only at the 16-to-32-cell stages of development in mesendoderm lineages. The on-going expression of these factors in different lineages does not preclude their role in NNE cell specification, but rather suggests pleiotropic requirements. Indeed, in order to unravel the role of NNE factors at the 16-cell stage from their later roles, we focused on the effect of their loss at the 32-cell stage, that is one cell cycle after the onset of NNE factor expression. The 32-cell stage corresponds to the stage when the lineage segregation of notochord-neural (NN) and endoderm (E) takes place. Later stages were also addressed, in part to tie in our data with the published literature. What our data showed was that all three factors were required individually for the correct initiation of *both* NN and E cell genetic programmes. This concern prompted us to highlight that these factors likely play on-going roles during NN and E lineage specification.

We also disagree that the NNE cells are an example of multi-lineage priming. We found no evidence for this. If NNE cell fate specification were a case of multi-lineage priming, one would predict that loss of individual NNE factors (Foxa.a, Foxd or *Fgf9*/16/20) would result in loss of *either* NN *or* E gene expression. However, (initially to our surprise) both NN and E gene expression was reduced if any one of these factors was inhibited. We concluded from this that NNE factors are not indicative of a primed state. In support of this proposition, overexpression of any one factor in developing ectoderm lineages was unable to promote ectopic mesoderm fate, whereas overexpression of all three factors could. This is again not what would be predicted if NNE cells were primed with a mixture of NN and E specifiers: in this case overexpression of individual NNE factors would promote either NN fates (for the NN factors) or E fates (for the E factors). In our system, it is clearly β-catenin and not cross-antagonism of primed NN and E factors, which generates the differential fate decision between NN and E cells.

Issues raised by both reviewers made it clear to us that we had not been sufficiently clear with what we meant by a ‘distinct’ regulatory state and how this differed from a ‘mixed' or 'mixed-lineage’ regulatory state. A ‘mixed-lineage’ regulatory state shares characteristics to a primed regulatory state whereby a mixture of cell-type specific genes are transcriptionally activated prior to their restricted expression in a cell-type specific fashion following lineage segregation (e.g. Graf and Enver, 2009, Nature). To put it more simply, a cell ‘AB’ expresses factors that are, individually, required for *either* of the subsequent fates ‘A’ *or* ‘B’. We preferred, however, not to call this a ‘primed’ regulatory state, since in many cases priming involves lows levels of gene activation followed by reinforcement in a lineage specific manner. Instead, we used the term ‘mixed-lineage regulatory state’ a term previously employed (Guo et al., Dev. Cell, 2010). A ‘distinct’ regulatory state would be, in contrast, a situation wherein the cell ‘AB’ expresses factors (AB lineage specification factors) and each of them is required for *both* subsequent states A *and* B. We realised that ‘distinct’ was perhaps not the best word, since a ‘mixed-lineage’ regulatory state is also distinct from either of the subsequent lineage-specific states. In the revised manuscript, we decided to avoid the use of the term ‘distinct regulatory state' and instead compare and contrast the regulatory situation of ascidians to that described in other systems. Our manuscript now includes a clear definition of a ‘mixed-lineage regulatory state’ with supporting schematics (Figure 6). We hope that the schematics more clearly highlight the difference between regulatory architectures of ascidian mesendoderm regulatory state and a mixed-lineage regulatory state. We also decided to no longer bring this point up in the Introduction, but rather to focus this, now more in depth comparison, to the Discussion section.

To summarise our findings, we identified in *Ciona* the mesendoderm lineage specifiers, whose combinatorial activity is sufficient to reprogramme developing ectoderm cells to a mesendoderm state. This ectopic mesendoderm state segregates further into either mesoderm or endoderm lineages depending on β-catenin activity, in a similar way to the endogenous mesendoderm lineage.

Finally, by the authors' admission, several of these regulatory linkages have been documented before, so it is not always clear what the novel insights really are.

See response to the first comment.

*Reviewer #2:*

*In this manuscript, the authors make substantial progress in delineating the transcriptional circuitry underlying mesendoderm lineage specification. Through an extensive and well-documented series of manipulations the authors demonstrate that three transcription factors are both necessary and sufficient for establishing the tunicate mesendodermal lineage. The authors then attempt to employ their results to help resolve questions about the nature and evolution of mesendodermal regulatory states in various clades. However, some of these interpretations are unclear and potentially inaccurate as discussed below.*

*My primary concern regards the overall framework the authors used to interpret their results. In the Abstract, the authors propose the mesendoderm may represent an "obligatory" or "essential" developmental state and indicate that their data supports this assertion. However, it is not clear that this is an accurate assessment.*

According to this premise, the loss of the mesendoderm regulatory state should preclude the subsequent establishment of endoderm and mesodermal lineages. Yet, the authors’ data contradict this. Instead they found that loss of one or more key mesendodermal regulatory factors did not preclude the eventual establishment of endoderm or mesoderm. In particular, they showed that:

a) ZicL (a mesoderm lineage marker) expression was eventually restored after inhibition of FGF signaling (Figure 2);

b) Endoderm differentiation (assayed by alkaline phosphatase activity) occurred despite inhibition of FGF or FoxD (Figure 2).

We understand the reviewer’s concern. What we meant by ‘essential’ was ‘essential for normal specification’. Indeed, NN and E cells fail to correctly initiate their genetic programmes when *any one* of the three NNE factors is inhibited. We accept that perhaps our use of the word 'essential' was too enthusiastic since there is likely to be some regulative processes that lead to the recovery of tissues under certain conditions. Below, we respond to the particular points raised in the first concern and explain why we maintain that the NNE factors are required for correct mesendoderm formation.

Following Fgf9/16/20 inhibition, *Zic-r.b* expression is restored at the 64-cell stage in both notochord and neural precursors. However, this is not that surprising considering that while FGF-signalling is required for notochord fate, it is not compatible with neural fate (differential ERK activation between notochord and neural fates drives their binary fate choice). Thus, one would expect an FGF-independent expression of *Zic-r.b* at the 64-cell stage, at least in neural fated cells. Consistent with potentially independent gene regulation between the 32- and 64-cell stages, it has been shown that *Zic-r.b* expression at these stages can be mediated by separate enhancer elements (Anno et al., 2006, Genes Dev. Evol.). In the revised version of our manuscript, we described in more detail the regulation of *Zic-r.b* and the on-going role of FGF signals during the correct specification of the notochord lineage. While, it is not possible to ascertain whether FGF-signalling is absolutely required at each successive time point, we would nonetheless argue that FGF-signalling via ERK is involved during the step-by-step (*Zic-r.b* at 32-cell, *Bra* at 64-cell) specification of notochord.

In Foxd-MO or Fgf9/16/20-MO injected embryos, while endoderm gene expression is reduced, at least until the early gastrula stage, endoderm fate eventually recovers by larval stages. Thus, while we are confident that Foxd and Fgf9/16/20 are required for the *correct specification* of endoderm, we also agree that endoderm fate can recover. This kind of recovery is not unusual. In sea urchin for example, in the absence of *pmar1*, skeletogenic mesenchyme fate recovers as a result of the action of *blimp1*, a process coined ‘regulative recovery’ (Smith and Davidson, 2009; PNAS). To use an ascidian example, the loss of Fgf9/16/20 results in an early loss of the notochord genetic programme. However, notochord cells later recover to some extent via the action of Fgf8/17/18 (Yasuo and Hudson, 2007). To turn the argument around, we would not argue from this that Fgf9/16/20 is *not* required for notochord formation, it clearly is required for notochord cells to follow their correct developmental programme. In the current study, the data showing that loss of Foxd and Fgf9/16/20 in combination leads to a strong loss of endoderm supports a level of 'redundancy' between these two factors, but also a requirement. While we do not find the recovery of endoderm particularly worrying for our model, we do understand the reviewer’s concern. We have now been more careful to highlight our observation that the NNE factors are required for the correct initiation of NN and E lineages, with some recovery during later development. We also no longer refer to an 'essential' regulatory state.

*They also found that single "mesendoderm" factors were able to restore either mesoderm or endoderm even though were not able to restore the mesendoderm state, effectively bypassing this state and demonstrating that it is not essential or obligatory. In particular they showed that:*

FoxD alone was able to restore expression of Mesoderm lineage genes after loss of the mesendoderm regulatory state through disruption of B-catenin (Figure 4).

No, this is not correct. *Foxd* is not alone in this experiment. As shown in Figure 1, loss of β-catenin results in quite broad expression of *Foxa.a* in both a-line cells as well as A-line cells (which are now most likely converted to an a-line fate based on co-expression of *Foxa.a* and *Efna.d*(Figure 2)). On the other hand, *Foxd* injection results in loss of *Efna.d* (a potent inhibitor of FGF-MEK-ERK-signals) and weak activation of *Fgf9/16/20* (Figure 3—figure supplement 1). Thus, in β-catenin-MO injected embryos, injection of *Foxd* into the egg restores the co-expression of all three factors by the 16-cell stage. In support of this hypothesis, treatment of β-catenin-MO+Foxd mRNA embryos with UO126 to inhibit FGF-signalling results in a similar reduction of *Zic-r.b* and *Bra* as is seen in control embryos. We agree that we did not explain this experiment very well in our previous manuscript and in response to this concern of Reviewer 2, we now explained this experiment in detail.

Finally, in support of our hypothesis that all three factors are required to induce an NNE-like state, our overexpression data clearly show that *all three* factors are required to reprogramme ectoderm into mesendoderm (Figure 5).

*There are at least three ways that the authors could address these concerns.*

1) Clarify what they mean by an "essential" role for the mesendoderm and how their experimental results support this hypothesis.

We agree with Reviewer 2 that this was not sufficiently clear in our submitted manuscript. In response to both reviewers, we now include an explicit description of different regulatory architectures.

2) Modify their hypothesis, possibly by suggesting that the three mesendodermal factors are locked into a recursive regulatory relationship. This caveat might explain why disruption of one factor only temporarily blocks the establishment of this regulatory state (Figure 2). It may also serve to explain why expression of a single factor is sufficient to re-establish the mesendodermal state/subsequent germ layer sub-specification (Figure 3). If they do propose this explanation, they should also evaluate whether it is supported by any data.

We think that the reviewer is suggesting that the three NNE factors may be locked into a kind of regulatory ‘kernel’ whereby each factor induces the expression of the others? This is a nice idea, but it is not supported by our data. Inhibition of any one factor does not affect expression of any other (Figure 2—figure supplement 1), although injection of *Foxd* does induce low levels of Fgf9/16/20 expression. Most importantly for our conclusions, expression of a single factor is *not* sufficient to re-establish NNE fate in developing ectoderm cells, as described above.

*3) Discard their current hypotheses and focus on an alternative framework. For example, they could choose to focus their interpretation on different regulatory architectures that might underlie mesendoderm regulatory states in different embryos and some potential functional implications of these different architectures. This framework may allow a more productive comparison to other characterized endomesodermal circuits (nematode/sea urchins) as further discussed in the next point.*

*A secondary concern focuses on the authors' statement that a "distinct mesendodermal state" is not present in sea urchins (Introduction, first paragraph and Discussion, second paragraph). The sea urchin and sea star GRNs both include a conserved, transient mesendodermal state based on B-catenin dependent expression of wnt8 and Otx (Hinman, Yankura and McCauley 2009). The cited paper (Peter and Davidson, 2010) does state that:*

*"In a strict sense, early endomesoderm specification therefore involves the concomitant activation of two separate GRNs, rather than the activation of a common endomesoderm GRN which diverges into two daughter programs. The co-expression of regulatory genes to which we refer to as endomesoderm regulatory state is the result of these two GRNs."*

*However, this interpretation is not definitive and a good argument could be made that the endomesodermal state is established prior to the simultaneous activation of the two germ-layer specific programs, as laid out in the cited Hinman paper. The authors should clarify exactly what they think defines a distinct endodermal regulatory state and whether or not their data suggests a tunicate regulatory circuitry that is fundamentally distinct from that observed in sea urchins/sea stars.*

Although we did not frame the question specifically in these terms, we were indeed trying to compare the architecture of the initial mesendoderm GRN of ascidians to that of different embryos. The specific aim of our study was to identify the mesendoderm lineage specifiers of the ascidian embryo and reveal how they influence the outcome of the second β-catenin driven binary fate decision that segregates the NN and E lineages. We concluded that ascidian mesendoderm bears most similarity to that in the *C.elegans* model and is not based on a mixed-lineage regulatory architecture. To date, to our knowledge, no mesendoderm lineage specifiers (that is individual factors at the top of the genetic hierarchy that are required for initiation of both mesoderm and endoderm GRNs) have been identified in other systems, other than the inducers (β-catenin and Nodal) themselves. However, the reviewer is correct to say that the published literature does not exclude the existence such mesendoderm specifiers upstream of both mesoderm and endoderm GRNs in sea urchins and vertebrates. It may simply be that regulative recovery/redundancy, together with extremely complicated early development and GRNs make the discovery of these kinds of factors very difficult. Or it may be that ascidians (and nematodes) do indeed use a fundamentally distinct mode of mesendoderm specification due to the simplification and rapidity of their early development. In the revised version of our manuscript, we attempt to address all of these issues in more detail.